# Brief communication: Impact of common ice mask in surface mass balance estimates over the Antarctic ice sheet

Nicolaj Hansen[1,2], Sebastian B. Simonsen[2], Fredrik Boberg[1], Christoph Kittel[3], Andrew Orr[4],
Niels Souverijns[5,6], J.Melchior van Wessem[7], and Ruth Mottram[1]

[1] Danish Meteorological Institute, Copenhagen, Denmark
[2] Geodesy and Earth Observation, DTU-Space, Technical University of Denmark, Lyngby, Denmark
[3] Laboratory of Climatology, Department of Geography, SPHERES, University of Liège, Liège, Belgium
[4] British Antarctic Survey, High Cross, Madingley Road, Cambridge, UK
[5] Department of Earth and Environmental Sciences, KU Leuven, Belgium
[6] Environmental Modelling Unit, Flemish Institute for Technological Research (VITO), Mol, Belgium
[7] Institute for Marine and Atmospheric Research Utrecht, Utrecht University, Utrecht, the Netherlands

**Correspondence:** Nicolaj Hansen (nichsen@space.dtu.dk)

**Abstract.** Regional climate models compute ice sheet surface mass balance (SMB) over a mask that defines the area covered by glacier ice, but ice masks have not been harmonised between models. Intercomparison studies of modelled SMB therefore use a common ice mask. The SMB in areas outside the common ice mask, which are typically coastal and high precipitation regions, are discarded. Ice mask differences change integrated SMB by between 40.5 to 140.6 Gt yr$^{-1}$, (1.8% to 6.0% of ensemble mean SMB), equivalent to the entire Antarctic mass imbalance. We conclude there is a pressing need for a common ice mask protocol.

## 1 Introduction

Detailed estimates of the surface mass balance (SMB) of the Antarctic ice sheet (AIS) are important for interpreting observed ice and sea-level rise budgets. SMB is the difference between accumulation and ablation at the surface of the ice sheet, which in Antarctica is positive due to the precipitation term dominating, especially in coastal areas, where high relief is often found due to complex/steep orography leading to orographic precipitation (Lenaerts et al., 2019).

Multiple regional climate models (RCMs) are now used to provide estimates of present-day and projected SMB. A selection of five of these were recently the subject of an intercomparison exercise (Mottram et al., 2021) to explore their commonalities and differences. Modelled present-day SMB for the total AIS (ToAIS, which we define as the ice sheet including ice shelves) in the scientific literature ranges from 2177±80 Gt yr$^{-1}$ to 2583 ±122 Gt yr$^{-1}$ (van Wessem et al., 2018; Souverijns et al., 2019; Agosta et al., 2019; Hansen et al., 2021). The intercomparison in Mottram et al. (2021) gives an ensemble mean of 2329±94Gt yr$^{-1}$ for the common period of 1987 to 2015 with a range of 1961±70 to 2519±118 Gt yr$^{-1}$ for individual models. Their results also show that while all models vary on an interannual basis directed by the driving ERA-Interim reanalysis, the spread in mean annual SMB estimates originates predominantly from differences in the dynamical core, physical parameterisations, model set-up and the Digital Elevation Model (DEM). Spatial differences in the pattern of SMB can be attributed locally to resolution and

differences in orography, but continental-scale variability in the distribution of precipitation is related predominantly to model physics and dynamical cores (Mottram et al., 2021). While all models were able to reproduce observed temperatures, pressures and wind speeds measured at automatic weather stations across the continent, models with nudging or daily reinitialisation had in general smaller biases, but even so differing regional patterns of SMB (see e.g. figure 4 in Mottram et al. (2021)).

Comparisons with the limited SMB observations available show that different models perform better in different regions and at different altitudes, making it challenging to draw general conclusions on a continental scale. Although there are multiple reasons for the range in estimates of SMB, we focus solely on the ice masks in this study in order to assess differences in SMB introduced in post-processing of model estimates. The Mottram et al. (2021) intercomparison study used a common ice mask to remove continent-wide present-day SMB differences related to variations in native ice mask extent. Ice masks are typically

made up of a binary grid that defines ice-covered areas, including ice shelves, and the ocean.

Here, we aim to quantify the importance of the ice mask in explaining the difference in SMB simulated by the five RCMs used in the Mottram et al. (2021) intercomparison study: COSMO-CLM[2] (Souverijns et al., 2019), HIRHAM5 (Hansen et al., 2021), MARv3.10 (Agosta et al., 2019), MetUM (Orr et al., 2015) and RACMO2.3p2 (van Wessem et al., 2018). We investigate the importance of the different native ice masks by creating a surface categorization that shows the number of models that are

35 represented in each grid cell. Furthermore, we show the spatial differences on basin-scale induced by the common mask. All the native masks have been regridded onto the same grid at $0.11°(\approx 12.5$ km) resolution.

Usually, SMB estimates over the AIS are confined to the grounded AIS, because it is only the mass change over the grounded AIS that results directly into sea-level change (Lenaerts et al., 2019). However, we include the ice shelves in all results, due to their buttressing of the main ice sheet and thereby importance for the general ice sheet dynamics (Dupont and Alley, 2005). We

also show the integrated $\Delta$SMB for the grounded ice sheet in Table 1. Thinning of the ice shelves has already been observed, which results in less buttressing and increased discharge from the grounded ice into the ocean (Gudmundsson et al., 2019). Further, the largest change in end of century projected AIS surface mass balance is shown to occur over the ice shelves (Kittel et al., 2021), it is therefore an important feature to get right in Antarctic modelling.

## 2 Methods

The five models were run in the Antarctic domain from 1987 to 2015 at different horizontal spatial resolutions and different land-sea mask data-sets: COSMO-CLM at 0.22°resolution, with ice mask created with data from the Scientific Committee on Antarctic Research (SCAR) Antarctic Digital Database version 5.0 published in 2006 (Lawrence et al., 2019); HIRHAM5 at two resolutions of 0.11°and 0.44°, with an ice mask derived from data created by the United States Geological Survey (USGS) Earth Resources Observation and Science (EROS) Center and consist of Advanced Very High Resolution Radiometer

(AVHRR) data in 1 km resolution collected from 1992 to 1993 (Eidenshink and Faudeen, 1994); MetUM at 0.44°resolution, has an ice mask created from the International Geosphere-Biosphere Programme (IGBP) data in combination with 1 km AVHRR from the period of 1992 to 1993 (Loveland et al., 2000); MARv3.10 at 0.32°resolution, with ice mask created from Bedmap2, which consist of a combination of different data sources such as satellite images, radar and laser altimetry gathered between

2000 and 2010 (Fretwell et al., 2013); and RACMO2.3p2 at 0.25°resolution, with ice mask made from a 1 km DEM that was
created from the combination of ERS-1 data from 1994 and ICESat data from 2003 to 2008 (Bamber et al., 2009).

Mottram et al. (2021) regridded all native ice masks and SMB estimates to a common grid of 0.11°. We adopt this approach, and refer to Mottram et al. (2021) for details. We then compare the SMB over the common mask and the native ice masks used in each individual original model simulation. The common mask is defined as all points where all the regridded native ice masks have grid cells that are covered with permanent ice. We break the simulated SMB down to the basin-scale using Antarctic drainage basins derived from Zwally et al. (2012), including ice shelves (see Fig. 1). Note that the Zwally et al. (2012) basins define an outer edge of the ToAIS, in all cases the native masks are slightly larger than the Zwally et al. (2012) definition. To make the grounded ice sheet values in table 1, we derived the values of the grounded ice sheet, using the data set for grounded ice from Zwally et al. (2012), since not all the native masks explicitly distinguished grounded from floating ice. This way we insured all the native masks and the common mask were compared equally. In table 2 we opted to define differences only over the Zwally defined basins, in order to be consistent with other studies. Both the area and the SMB have been calculated for the common mask and native masks in each of the 27 basins. Three values are given for each model basin: $\Delta area_\%$ the percentage difference in area, $\Delta SMB_\%$ the percentage difference in SMB, and $\Delta SMB_{Gtyr^{-1}}$ the difference in SMB in Gt yr$^{-1}$. All calculations are derived by subtracting the regridded native mask from the common mask ($\Delta = common - native$).

## 3 Results

Comparing the area of the common mask to the area of the native masks, we see that the common mask is between 1.85% and 2.89% smaller than the different native masks, Table 1. This results in integrated SMB values that are between 40.5 and 140.6 Gt yr$^{-1}$ smaller when using the common mask compared to the native mask, which is up to 6.04% of the ensemble mean SMB (Mottram et al., 2021), Table 1. The two HIRHAM5 simulations are very close to having identical areas, however the SMB is larger in the 0.11°simulation, which is most likely due to the steep coastal orography being better resolved and thus leading to an increased orographic precipitation (Webster et al., 2008). In the rightmost column the $\Delta SMB_{Gtyr^{-1}}$ over the grounded AIS is shown, when excluding the ice shelves there are still differences between the common mask and the native masks of between 20.1 to 102.4 Gt yr$^{-1}$, Table 1. It should be pointed out that a small $\Delta$ value is not necessarily more correct than a large $\Delta$ value in terms of true area size or SMB magnitude. It solely refers to how close it is to the common mask and the ensemble mean from Mottram et al. (2021).

**Table 1.** From left to right the columns show: RCM name, $\Delta$area$_\%$ and $\Delta$SMB$_{\text{Gtyr}^{-1}}$ both between the common mask and the native mask over the ToAIS, $\Delta$SMB$_\%$ is the difference between the $\Delta$SMB$_{\text{Gtyr}^{-1}}$ and the SMB ensemble mean from Mottram et al. (2021) which is 2329 Gt yr$^{-1}$, the yearly SMB for the individual models integrated over the common mask with uncertainties of one standard deviation, and finally the $\Delta$SMB$_{\text{Gtyr}^{-1}}$ over the Grounded AIS.

| Model | $\Delta$area [%] | $\Delta$SMB [Gt yr$^{-1}$] | $\Delta$SMB [%] | SMB [Gt yr$^{-1}$] | Grounded $\Delta$SMB [Gt yr$^{-1}$] |
|---|---|---|---|---|---|
| HIRHAM5 0.11° | -2.43 | -140.6 | -6.04 | 2452±107 | -102.4 |
| HIRHAM5 0.44° | -2.49 | -69.5 | -2.99 | 2518±118 | -40.7 |
| MARv3.10 | -2.89 | -91.9 | -3.95 | 2445±91 | -54.1 |
| COSMO-CLM$^2$ | -1.94 | -40.5 | -1.77 | 1961±70 | -20.1 |
| RACMO2.3p2 | -1.85 | -119.6 | -5.13 | 2399±101 | -74.0 |
| MetUM | -2.49 | -57.6 | -2.47 | 2191±101 | - 33.9 |

The different model-mask combinations are shown in Figure 1. Around the Antarctic Peninsula (basins 24-27) there are large mask disagreements over Larsen C ice shelf, at the tip of the Peninsula and the surrounding islands. Parts of west AIS, especially glaciers such as Getz (basin 20), Thwaites (basin 21) and Abbot (basin 23) also have large mask disagreements, in East AIS it the places like Fimbul, Amery and West ice shelves, Fig. 1. All around the coastline we see, going from the ice sheet and out towards the ocean, that the number of ice masks outside the common mask decreases. Furthermore, two of the masks, COSMO-CLM$^2$ and MARv3.10, contain non-iced grid cells in their native masks, simulating some parts of non-ice covered parts of the Transantarctic Mountains, Fig. 1 panel A. Moreover, it is shown how well the common mask agrees to a newly derived ice mask. We compare the common mask to the Reference Elevation Model of Antarctica (REMA, Howat et al. (2019)) mask over the Antarctic Peninsula, here it is clear that the common mask is smaller than the REMA in most coastal areas around the AP, Fig. 1 panel B.

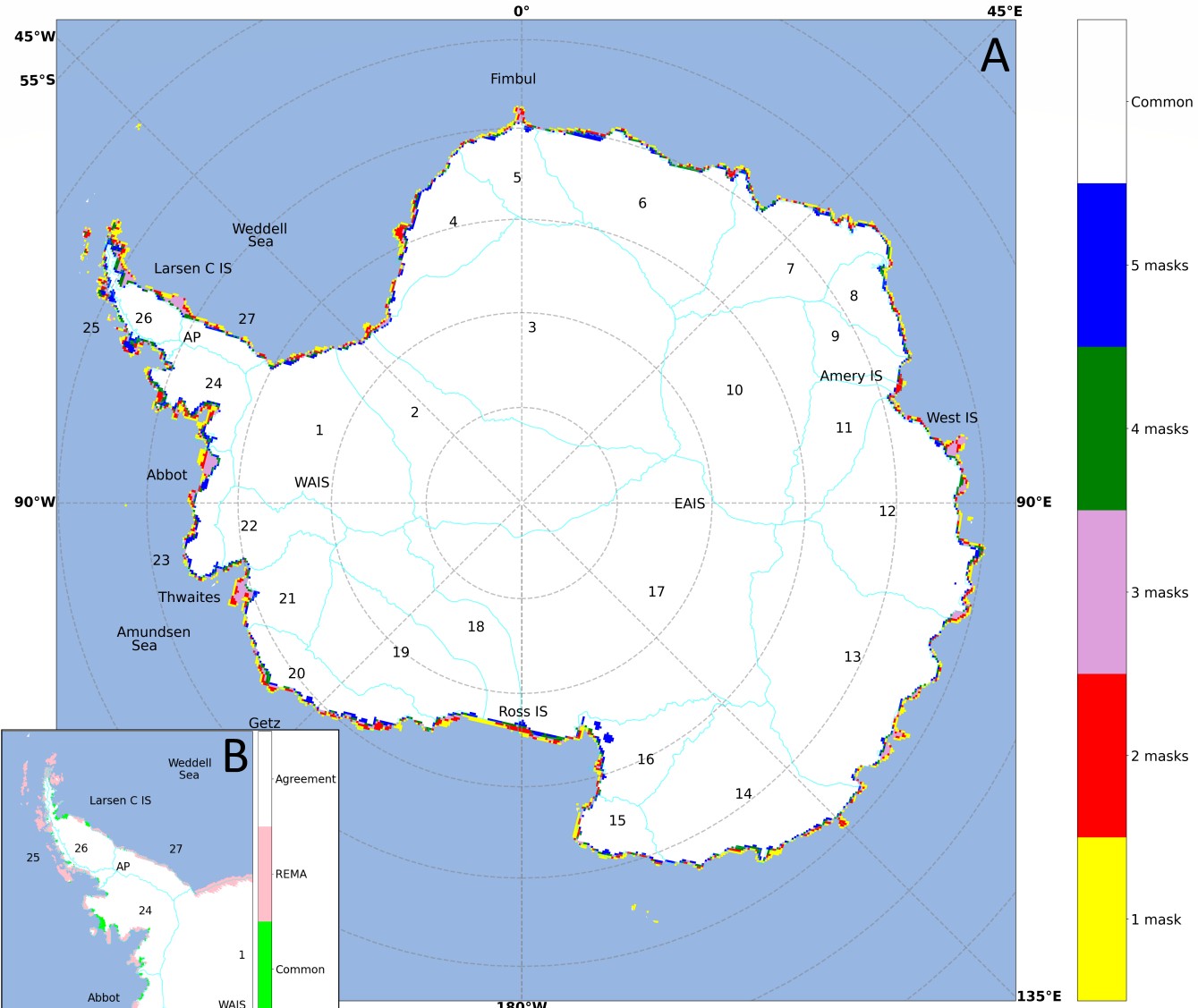

**Figure 1.** Panel A shows the ice mask agreement, the white area is the common mask (i.e where all six masks have ice), the colours around the coastline represents the number of native masks that have ice outside the common mask. The numbers refer to the 27 drainage basins with ice shelves, outlined in turquoise, and "IS" is short for ice shelf and AP, WAIS and EAIS is the Antarctic Peninsula, west AIS and east AIS respectively. Panel B is a cut-of, of the AP display the mask of the REMA mask, the common mask, and where they agree.

In order to investigate the regional and basin-scale variability, Table 2 shows values of $\Delta area_\%$, $\Delta SMB_\%$ and $\Delta SMB_{Gtyr^{-1}}$ for each of the models and for each basin. $\Delta SMB_{Gtyr^{-1}}$ for basins 20, 23, 24 and 25 is the most sensitive to changes in the mask definition, up to 42.2 Gt yr$^{-1}$. Of these four, two basins (20 and 23) are in West Antarctica and two on the windward side of the Antarctic Peninsula (24 and 25). Furthermore, the relative difference between $\Delta area_\%$ and $\Delta SMB_\%$ in the model-basin

combinations shows a large variability between the models and between the basins (Table 2). Summed over all the basins
COSMO-CLM$^2$ has the smallest relative difference between $\Delta$area$_\%$ and $\Delta$SMB$_\%$. COSMO-CLM$^2$ also has the smallest
$\Delta$SMB$_{Gtyr^{-1}}$ integrated over the 27 basins, which show that COSMO-CLM$^2$ is least affected by the change in ice mask, Table
1. Examination of Mottram et al. (2021) shows that COSMO-CLM$^2$ is the driest model in the intercomparison and HIRHAM5
0.44° is the wettest, followed by HIRHAM5 0.11° and RACMO2.3p2. All six model simulations show differences between 0
and -2 Gt yr$^{-1}$ in $\Delta$SMB$_{Gtyr^{-1}}$ in basins 2 and 3, which have outlet to the Weddell sea, basins 8, 9, 10 and 11 surrounding the
Amery ice shelf, basins 16, 17, 18, 19 surrounding the Ross ice shelf, basin 22 with outlet in the Amundsen sea and basin 27
on the lee side of the Antarctic Peninsula. Of these 12 basins, 8 of them are in East Antarctica.

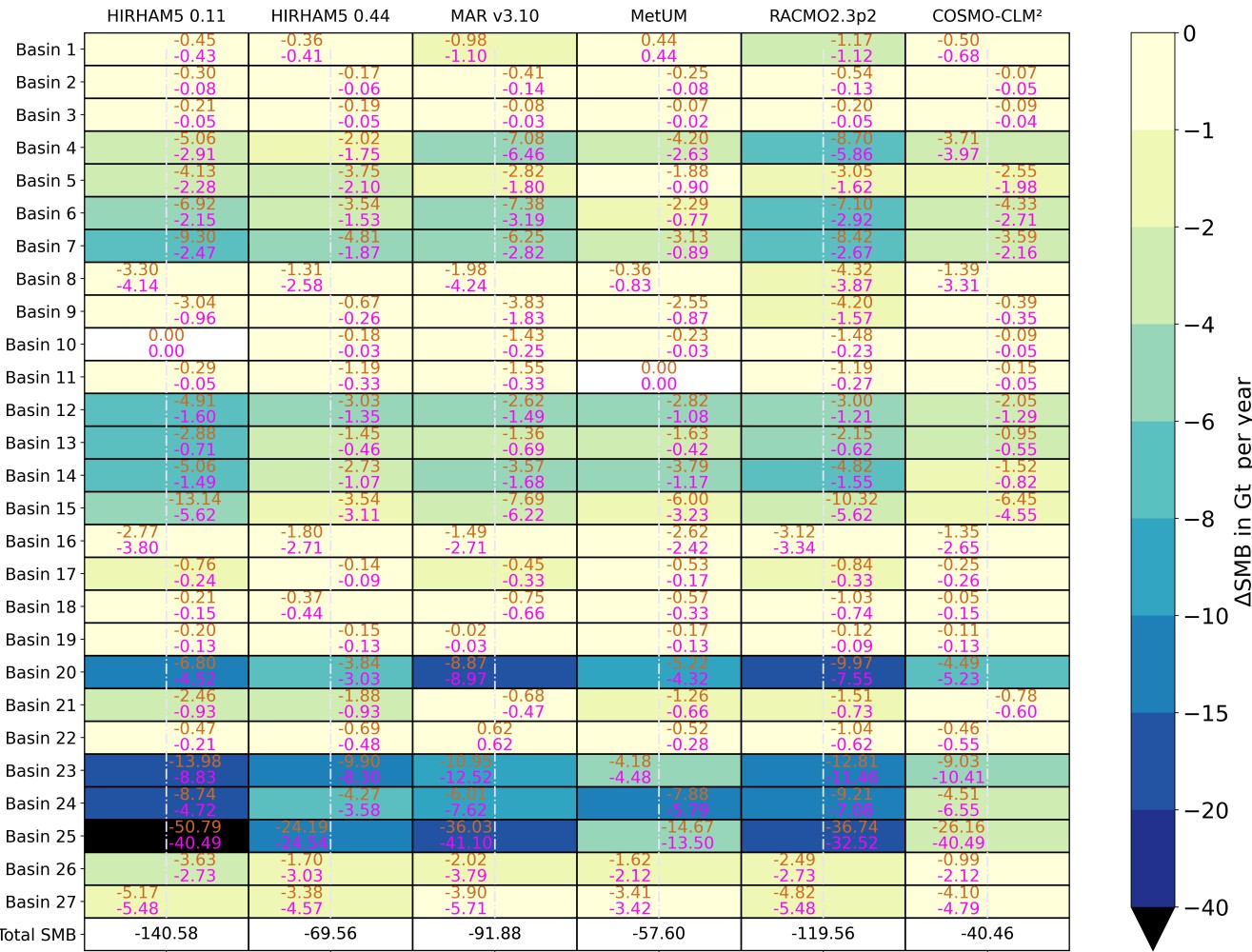

| | HIRHAM5 0.11 | HIRHAM5 0.44 | MAR v3.10 | MetUM | RACMO2.3p2 | COSMO-CLM² |
|---|---|---|---|---|---|---|
| Basin 1 | -0.45 / -0.43 | -0.36 / -0.41 | -0.98 / -1.10 | 0.44 / 0.44 | -1.17 / -1.12 | -0.50 / -0.68 |
| Basin 2 | -0.30 / -0.08 | -0.17 / -0.06 | -0.41 / -0.14 | -0.25 / -0.08 | -0.54 / -0.13 | -0.07 / -0.05 |
| Basin 3 | -0.21 / -0.05 | -0.19 / -0.05 | -0.08 / -0.03 | -0.07 / -0.02 | -0.20 / -0.05 | -0.09 / -0.04 |
| Basin 4 | -5.06 / -2.91 | -2.02 / -1.75 | -7.08 / -6.46 | -4.20 / -2.63 | -8.70 / -5.86 | -3.71 / -3.97 |
| Basin 5 | -4.13 / -2.28 | -3.75 / -2.10 | -2.82 / -1.80 | -1.88 / -0.90 | -3.05 / -1.62 | -2.55 / -1.98 |
| Basin 6 | -6.92 / -2.15 | -3.54 / -1.53 | -7.38 / -3.19 | -2.29 / -0.77 | -7.10 / -2.92 | -4.33 / -2.71 |
| Basin 7 | -9.30 / -2.47 | -4.81 / -1.87 | -6.25 / -2.82 | -3.13 / -0.89 | -8.42 / -2.67 | -3.59 / -2.16 |
| Basin 8 | -3.30 / -4.14 | -1.31 / -2.58 | -1.98 / -4.24 | -0.36 / -0.83 | -4.32 / -3.87 | -1.39 / -3.31 |
| Basin 9 | -3.04 / -0.96 | -0.67 / -0.26 | -3.83 / -1.83 | -2.55 / -0.87 | -4.20 / -1.57 | -0.39 / -0.35 |
| Basin 10 | 0.00 / 0.00 | -0.18 / -0.03 | -1.43 / -0.25 | -0.23 / -0.03 | -1.48 / -0.23 | -0.09 / -0.05 |
| Basin 11 | -0.29 / -0.05 | -1.19 / -0.33 | -1.55 / -0.33 | 0.00 / 0.00 | -1.19 / -0.27 | -0.15 / -0.05 |
| Basin 12 | -4.91 / -1.60 | -3.03 / -1.35 | -2.62 / -1.49 | -2.82 / -1.08 | -3.00 / -1.21 | -2.05 / -1.29 |
| Basin 13 | -2.88 / -0.71 | -1.45 / -0.46 | -1.36 / -0.69 | -1.63 / -0.42 | -2.15 / -0.62 | -0.95 / -0.55 |
| Basin 14 | -5.06 / -1.49 | -2.73 / -1.07 | -3.57 / -1.68 | -3.79 / -1.17 | -4.82 / -1.55 | -1.52 / -0.82 |
| Basin 15 | -13.14 / -5.62 | -3.54 / -3.11 | -7.69 / -6.22 | -6.00 / -3.23 | -10.32 / -5.62 | -6.45 / -4.55 |
| Basin 16 | -2.77 / -3.80 | -1.80 / -2.71 | -1.49 / -2.71 | -2.62 / -2.42 | -3.12 / -3.34 | -1.35 / -2.65 |
| Basin 17 | -0.76 / -0.24 | -0.14 / -0.09 | -0.45 / -0.33 | -0.53 / -0.17 | -0.84 / -0.33 | -0.25 / -0.26 |
| Basin 18 | -0.21 / -0.15 | -0.37 / -0.44 | -0.75 / -0.66 | -0.57 / -0.33 | -1.03 / -0.74 | -0.05 / -0.15 |
| Basin 19 | -0.20 / -0.13 | -0.15 / -0.13 | -0.02 / -0.03 | -0.17 / -0.13 | -0.12 / -0.09 | -0.11 / -0.13 |
| Basin 20 | -6.80 / -8.52 | -3.84 / -3.03 | -8.87 / -8.97 | -5.22 / -4.32 | -9.97 / -7.55 | -4.49 / -5.23 |
| Basin 21 | -2.46 / -0.93 | -1.88 / -0.93 | -0.68 / -0.47 | -1.26 / -0.66 | -1.51 / -0.73 | -0.78 / -0.60 |
| Basin 22 | -0.47 / -0.21 | -0.69 / -0.48 | 0.62 / 0.62 | -0.52 / -0.28 | -1.04 / -0.62 | -0.46 / -0.55 |
| Basin 23 | -13.98 / -8.83 | -9.90 / -8.30 | -10.95 / -12.52 | -4.18 / -4.48 | -12.81 / -11.46 | -9.03 / -10.41 |
| Basin 24 | -8.74 / -4.72 | -4.27 / -3.58 | -6.81 / -7.62 | -7.88 / -5.79 | -9.21 / -7.08 | -4.51 / -6.55 |
| Basin 25 | -50.79 / -40.49 | -24.19 / -24.41 | -36.03 / -41.10 | -14.67 / -13.50 | -36.74 / -32.52 | -26.16 / -40.49 |
| Basin 26 | -3.63 / -2.73 | -1.70 / -3.03 | -2.02 / -3.79 | -1.62 / -2.12 | -2.49 / -2.73 | -0.99 / -2.12 |
| Basin 27 | -5.17 / -5.48 | -3.38 / -4.57 | -3.90 / -5.71 | -3.41 / -3.42 | -4.82 / -5.48 | -4.10 / -4.79 |
| Total SMB | -140.58 | -69.56 | -91.88 | -57.60 | -119.56 | -40.46 |

**Table 2.** Rows 1 to 27 are basins and the columns are the RCM's. There are two numbers in each grid cell, the bottom number in magenta is the difference in area between the common mask and native masks ($\Delta area_\%$) in each basin. The upper number in orange show difference in SMB between the common mask and native masks ($\Delta SMB_\%$) in each basin. The grid cell colour shows the difference in SMB in Gt yr$^{-1}$ ($\Delta SMB_{Gtyr^{-1}}$). All calculations are derived by subtracting the native mask from the common mask ($\Delta = common - native$). The differences are shifted left if $\Delta area_\%$ is greater than $\Delta SMB_\%$ and right if $\Delta SMB_\%$ is greater than $\Delta area_\%$. The bottom row is the summed up $\Delta SMB$ values over over 27 basins, for each models.

# 4 Discussion

We find the differences between common and native ice mask areas small (<3%), but it alters the SMB by up to 6% over the ToAIS (140.6 Gt yr$^{-1}$) when compared to the ensemble mean from Mottram et al. (2021). RACMO2.3p2, MARv3.10 and HIRHAM5 0.11°all have $\Delta SMB_{Gtyr^{-1}}$ values close to or larger than their given uncertainties for their respective SMB estimate.

This means that the effect of using the common mask in estimating SMB is close to or greater than the standard deviation of annual mean SMB estimates derived from the interannual variability in modelled SMB. We consider the standard deviation to be a minimum estimate of uncertainty within each model with actual uncertainties likely to be considerably larger, but difficult to estimate accurately (Lenaerts et al., 2019). Over the grounded AIS the common mask alters the SMB by up to 102 Gt yr$^{-1}$, see Table 1. This difference in SMB is close in magnitude to the grounded AIS mass loss of 109$\pm$56 Gt yr$^{-1}$ between 1992 and 2017 determined by the second ice sheet mass balance inter-comparison exercise (IMBIE2, Shepherd et al. (2018)), and thereby essentially determining if the AIS is losing or gaining mass. This means that small changes in SMB can lead to a non-negligible change in the total mass budget of the AIS. The model mean of the grounded $\Delta SMB_{Gtyr^{-1}}$ is 54.2 Gt yr$^{-1}$, which would make a sizeable change in the mass balance results, Table 1. Basin 25 has few or no ice shelves, thus it has one of the largest impacts for $\Delta SMB_{Gtyr^{-1}}$ for both the grounded basin (not shown) and when ice shelves are included.

Given the importance of the ice shelves to the dynamics of grounded ice, we argue that they are important to include in SMB models accurately. Furthermore, we speculate that there may be similar considerations when defining the grounded ice sheet for SMB assessments. This clearly shows the RCMs ice mask is key in integrated assessments of the Antarctic mass balance. These differences between area change and SMB change are the result of the area differences being located in the coastal regions, some of which are also high relief regions, leading to effects on the SMB that are disproportionately high relative to the area.

In each grid cell between 1 and 6 native masks can have ice in it, this results in 63 combinations of model coverage, in addition to the common mask (not shown). These different area coverage combinations around the ice sheet are partly driven by differences in ice masks and partly by differences in resolution. However, we cannot identify any systematic or model-specific biases on a regional scale. The native ice masks vary around the coastline arbitrarily, which is partly due to the time when the ice mask was created and what data are used to create the native ice mask. For example, the HIRHAM5 and MetUM ice masks are created from data collected three decades ago, yet there have been multiple calving events since the data collection. The native ice mask from COSMO-CLM$^2$ and MARv3.10 are created from data collected more recently and in a higher resolution. The higher resolution is also a benefit over Antarctic Peninsula and in coastal areas where there is complex orography, where a higher resolution also can change the orographic precipitation.

The common mask is introduced during the post-processing stage after running the RCMs with their native masks. This has the disadvantage that model variables where the fluxes are linked to the orography, such as precipitation, can introduce a bias, if the native mask is located differently in the domain, compared to the common mask. The same orography bias can be true for winds and thus the sublimation rates as well. High precipitation rates are often strongly linked to the steep orography in coastal areas around Antarctica, especially in West Antarctica and on the windward side of the Antarctic Peninsula (basins 24 and 25), which is also where we see the largest differences in $\Delta SMB_{Gtyr^{-1}}$ in Table 2. Comparing the size of the common mask with the REMA mask over the Antarctic Peninsula shows that the common mask is smaller around most of the coastline, Fig. 2 panel B.

The large differences in $\Delta SMB_{Gtyr^{-1}}$ and $\Delta area_\%$ found in this study suggest the need for more work to be done on a community basis to define a common mask, ideally before further RCM runs are conducted with the aim of contributing to

any model intercomparison study. As the computational demands vary from model to model, we cannot expect the modelling groups to run on the same spatial grid resolution. Therefore, we suggest a three-staged effort to be undertaken: Step 1: Agree on a state-of-the-art DEM of the Antarctic continent, with a sufficiently high spatial resolution to be appropriate for even kilometre and sub-kilometre models (Orr et al., 2021). This could in our minds be the REMA (Howat et al., 2019) at 100-metre grid resolution. This 100-metre grid would form the basis for a community grid. Step 2: Agree on a state-of-the-art delineation of surface types, such as bare rock, ice shelves, ice sheet etc., for the Antarctic continent, again with the highest possible resolution. Here, we see the grid-independent delineation of Antarctic glaciers as provided by the Randolph Glacier Inventory's shapefiles (RGI Consortium, 2017), combined with the shapefiles from the High resolution vector polygons of the Antarctic coastline data-set (Gerrish et al., 2021) as a possibility. These two will then be combined to give the ice mask on the 100-metre grid. Step 3: Provide the community with a common tool for projecting the 100-metre grid onto the RCM modelling grid. We suggest that tool will as a minimum give the variables to be used in further model intercomparisons: grid area (we suggest the antarctic domain as defined in the Coordinated Regional Climate Downscaling Experiment (CORDEX)), surface elevation, surface types so one can distinguish between grounded ice, floating ice (glaciers/tongues) and rocks/mountains-ridges and ice-cover percentage. The ice-cover percentage will then provide the needed information to have the models contributing equally despite being run on different model resolutions, as the topography of the ice sheet is the same for all models. We imagine the tool consists of needed data in high resolution and a script that can create the grid file in the wanted resolution, possibly in a netCDF format, since most modeling groups are use to working in this format, and it will be easy to handle. Ideally the ice mask would evolve over time as the continent changes, however, we feel it is important to first standardize the approach to creating and using ice masks. Thus, we suggest that it is noted in the mask file, which data sets are used to create it, and when the data sets are last updated.

## 5 Conclusions

We have quantified the importance of the choice of ice mask for the Antarctic domain by comparing six different ice masks from the RCM; COSMO-CLM$^2$, HIRHAM5 (in two resolutions), MARv3.10, MetUM and RACMO2.3p2 with the common mask defined by Mottram et al. (2021). We find differences between 40.5 Gt yr$^{-1}$ and 140.6 Gt yr$^{-1}$ over the ToAIS and differences between 20.1 and 102.4 yr$^{-1}$ over the grounded AIS (Table 1), comparing the native mask to the common mask integrated over all the basins in Antarctica. Looking at individual basins, we find area differences from 0% (HIRHAM5 0.11°basin 10 and MetUM basin 11) up to 40.49% (HIRHAM5 0.11°basin 25) between the common and native masks, (Table 2). Furthermore, area changes do not map to SMB change linearly (Table 2). The biggest differences are in basins 20, 23, 24 and 25, showing that areas with high SMB are most sensitive to mask differences (Table 2). As the native masks are created from different data sets they do not all include the same ice shelves and ice tongues. We speculate, that we introduce a shift in value by first defining the common mask after RCM simulations have been performed on their native masks. Most of the model variables in the SMB equation are sensitive to the orography and therefore sensitive to the representation of it when integrating over the common

mask. The effort of defining a common mask should ideally be a community effort and should be done before conducting further model intercomparisons and contributing to joint assessments of mass balance such as the IMBIE assessment.

*Data availability.* The common and native grid estimates of SMB over native and common masks are available at
https://doi.org/10.11583/DTU.16438236.v1

*Author contributions.* NH, RM and SBS conceived the study and wrote the initial draft manuscript. Analysis of simulations was carried out by NH and SBS. Model simulation output and ice masks were provided by FB, CK, AO, JMVW, and NS. All authors revised and contributed to the final manuscript.

*Competing interests.* We declare no competing interests.

*Acknowledgements.* R. Mottram and F. Boberg acknowledge the support of the Danish State through the National Centre for Climate Research (NCKF). This publication was supported by PROTECT. This publication was supported by PROTECT. This project has received funding from the European Union's Horizon 2020 research and innovation programme under grant agreement No 869304, PROTECT contribution number 29. The COSMO-CLM2 integrations were supported by the Belgian Science Policy Office (BELSPO; grant no. 747 BR/143/A2/AEROCLOUD) and the Research Foundation Flanders (FWO; grant nos. 748 G0C2215N and GOF5318N; EOS ID: 30454083). Computational resources and services were provided by the Flemish Supercomputer Center, funded by the FWO and the Flemish Government, EWI department.

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
