# Peer review of "Brief communication: Impact of common ice mask in surface mass balance estimates over the Antarctic ice sheet"

_The Cryosphere, 2021_

## Author Comment (AC1)

Reply to reviewer comments on

**"Brief communication: Impact of common ice mask in surface mass balance estimates over the Antarctic ice sheet"**

by

Nicolaj Hansen, Sebastian B. Simonsen, Fredrik Boberg, Christoph Kittel, Andrew Orr, Niels Sourverijns, Melchoir van Wessen and Ruth Mottram

Dear Editor

On behalf of my co-authors and myself, I would like to thank the reviewers for their comments on our manuscript. The reviewers have made an extensive review of the manuscript language and we have followed their suggestions to our best efforts. In the following, we provide a point-by-point answer to all the issues raised by the reviewers. We have gathered and numbered all issues raised by the reviewers (Anonymous Referee #1: 1-24 and Anonymous Referee #2 25-31), all issues will be followed by our suggestions for improvement to the manuscript highlighted in red.

Best regards, Nicolaj Hansen

**Anonymous Referee #1**

 Understanding why there are differences between the ice masks seems relevant to investigate in this study. The authors spend a few lines in the beginning of the Methods on the background for each model, but we need to more information on the native ice masks, what are they made from? Do they have a time stamp? How to they deal with ice shelves, grounded ice? etc.? etc.? Asking (and answering ...) these questions and more provides a better understanding of "problems" with the native ice masks – being too old, old coarse, etc. Perhaps this can be used for more easily defining a concrete recommendation in addition to the 3 steps already defined? We have rewritten the beginning of the method section (Line 45-55 in the revised version):

"The five models were run in the Antarctic domain from 1987 to 2015 at different horizontal spatial resolutions and different land-sea mask data-sets: COSMO-CLM at 0.22 resolution, with ice mask created with data from the Scientific Committee on Antarctic Research (SCAR) Antarctic Digital Database version 5.0 published in 2006 (Lawrence et al., 2019); HIRHAM5at two resolutions of 0.11 and 0.44, with an ice mask derived from data created by the United States Geological Survey(USGS) Earth Resources Observation and Science (EROS) Center and consist of Advanced Very High Resolution Radiometer (AVHRR) data in 1 km resolution collected from 1992 to 1993 (EIDENSHINK and FAUNDEEN, 1994); MetUM at 0.44 resolution, has an ice mask created from the International Geosphere-Biosphere Programme (IGBP) data in combination with 1 km AVHRR from the period of 1992 to 1993 (Loveland et al., 2000); MARv3.10 at 0.32° resolution, with ice mask created from Bedmap2, which consist of a combination of different data sources such as satellite images, radar and laser altimetry gathered between 2000 and 2010 (Fretwell et al., 2013); and RACMO2.3p2 at 0.25° resolution, with ice mask made from a 1 km DEM that was created from the combination of ERS-1 data from 1994 and ICESat data from 2003 to 2008 (Bamber et al., 2009)."

2. There is a tendency when reading the manuscript that the "truth"/true AIS mask is the common mask. See for instance Table 1 and L 71 – "...smallest deltaSMB integrated over the 27 basins...". Why the comparison to the common ice mask? The answer is of course easy, it's what is common.

However, throughout the study there is an implied desire for deltaArea and deltaSMB to be as small as possible, but that does not make much sense as the extent of the common is likely some distance from real world observations. However, I understand that without a new ice mask/product ready to be used, the authors need something to compare against – and in reality, it is highlighting the differences between models that needs to be conveyed. A way of addressing this, and conveying the need for a common usable product, would be to investigate/digitize the coastline of a few selected basins where differences are both small and large. As the comparison is based on regridded data to 0.11 degr., the resolution should just need to be better that this for now. These areas of interest could be displayed as a new figure 2 or be added as panels to figure 1. The main text should include a paragraph outlining the problem between real world observations, ice masks of each model, and comparison to a common ice mask (due to the lack of a new...). This could be used to better convey why a comparison to the common mask is used for now. Following this, the remained of the text (wording/phasing) should be reassessed in this perspective.

To clarify a small delta is not necessarily more true, We have this sentence above Table 1 (Line 77-79 in the revised version):

"It should be pointed out that a small  $\Delta$  value is not necessarily more correct than a large  $\Delta$  value in terms of true area size or SMB magnitude. It solely refers to how close it is to the common mask and the ensemble mean from Mottram et al. (2021)"

Furthermore, we have added a comparison between the mask from the DEM REMA and the common mask over the Peninsula in figure 1, and added this text above figure 1 (Line 86-88 in the revised version):

"Moreover, it is also shown how well the common mask agreeswith the Reference Elevation Model of Antarctica (REMA, Howat et al. (2019)) mask over the AP, it is clear that the common mask is smaller than the REMA area, Fig. 1 panel B"

We have rewritten L71 to this (Line 94-95 in the revised version): "COSMO-CLM2 also has the smallest  $\triangle$ SMB Gtyr -1 integrated over the 27 basins, which show that COSMO-CLM2 is least affected by the change in ice mask, Tab 1"

3. Throughout there is a discussion on the delta-values, however, each model has an uncertain associated. To make an even stronger case I think the authors should include the SMB uncertainties in table 1 and use this as a basis for a discussion on delta vs uncertainty. As the authors mention in the

Introduction there is a range in the SMB values, but also in the uncertainties (listed 80 - > 122 gt/yr) equal to 3.7 % - 4.7%, which to me seems very, very low, in particular given the results of this study.

This is a very interesting comment and we agree with the reviewer that the uncertainties as quoted seem low. Uncertainty quantification in the true sense of the meaning of uncertainty on a modelled quantity compared to a measured quantity is extremely difficult for SMB calculated from models. The convention in the field is to use the standard deviation on SMB equating in effect to interannual variability to estimate uncertainty in SMB estimates. As this subject is worth a paper in its own right we use the standard deviation as calculated in Mottram et al and added these as uncertainties on the SMB in table 1, and added this to the discussion (Line 103-108 in the revised version):

"... ensemble mean from Mottram et al. (2021). RACMO2.3p2, MARv3.10 and HIRHAM5 0.11 • all have  $\Delta$ SMB Gtyr-1 values close to or larger than their given uncertainties for their respective SMB estimate. This means that the effect of using the common mask in estimating SMB is close to or greater than the standard deviation of annual mean SMB estimates derived from the interannual variability in modelled SMB. We consider the standard deviation to be a minimum estimate of uncertainty within each model with actual uncertainties likely to be considerably larger, but difficult to estimate accurately (Lenaerts et al 2019).

4. Also, this study builds on the publication by Mottram *et al.*, 2021, and although I am not a fan of adding unnecessary text for the sake of adding to the length, this manuscript could do with adding more on the findings/problems encountered by Mottram et al. Some of the suggestions below this.

We have modified and slightly expanded the paragraph to give a brief overview of the results from Mottram et al. It now reads (Line 12-129 in the revised version):

"Multiple regional climate models (RCMs) are now used to provide estimates of present-day and projected SMB and a selection of five of these were recently the subject of an intercomparison exercise (Mottram et al., 2021) to explore their commonalities and differences. Modelled present-day SMB for the total AIS (ToAIS, which we define as the ice sheet including ice shelves) in the scientific literature ranges from 2177±80 Gt yr-1to 2583±122 Gt yr-1(van Wessem et al., 2018; Souverijns et al., 2019; Agosta et al., 2019; Hansen et al., 2021). The intercomparison in Mottram et al. (2021) gives an ensemble

mean of 2329±94Gt yr-1 for the common period of 1987 to 2015 with a range of 1961±70 to 2519±118 Gt yr-1for individual models. Their results also show that while all models vary on an interannual basis directed by the driving ERA-Interim reanalysis, the spread in mean annual SMB estimates originates predominantly from differences in the dynamical core, physical parameterisations, model set-up and Digital Elevation Model (DEM). Spatial differences in the pattern of SMB can be attributed locally to resolution and differences in orography, but continental-scale variability in the distribution of precipitation is related pre-dominantly to model physics and dynamical cores (Mottram et al., 2021). While all models were able to reproduce observed temperatures, pressures and wind speeds measured at automatic weather stations across the continent, models with nudging or daily reinitialisation had in general smaller biases, but even so differing regional patterns of SMB (see e.g. figure 4 in Mottram et al. (2021)). Comparisons with the limited SMB observations available show that different models perform better in different regions and at different altitudes, making it challenging to draw general conclusions on a continental scale. Although there are multiple reasons for the range in estimates of SMB, we focus solely on the ice masks in this study in order to assess differences in SMB introduced in post-processing of model estimates. The Mottram et al. (2021) intercomparison study used a common ice mask to remove continent-wide present-day SMB differences related to variations in native ice mask extent."

5. L12: how do their anomalies compare?

We are unsure which anomalies the reviewer has in mind, however, we note that the RCMs in Mottram et al show very similar interannual variability derived from the driving model (ERA-Interim). See text in note 4 above.

6. L18-23: this is an example of where more info from Mottram2021 would be good, what are the key findings etc. Here we are dealing with area/ice mask, but what else cause differences?

See details under comment #4

7. L28: "in most of the result" why not all? Please ensure consistency throughout

Because we also show the grounded Delta SMB in the last rightmost column in table 1. We have rephrased the sentence to this (Line 38-40 in the revised version):

"However, we include the ice shelves in all results, due to their buttressing of the main ice sheet and thereby importance for the general ice sheet dynamics (Dupont and Alley,2005). We also show the integrated  $\triangle$ SMB for the grounded ice sheet in Tab. 1. Thinning of the ice ...... "

- 8. L28-34: Does this difference cause problems for the comparison? This kind of falls back on the description of the native grid of each model. We are unsure which differences the reviewer has in mind? Is it the difference between including/excluding the ice shelves? Then yes, it partly falls back on the native grid, where the differences are more pronounced when the ice shelves are included. However, we do not believe it causes a problem in the comparison, because we want to show these differences, and in Table 1 we have both the Delta SMB and Delta grounded SMB, thus we show both numbers and their differences.
- 9. L37 (and L26): how does re-gridding affect the results? Any differences for any of the models? Did you check for consistency and that no biases where introduced?

For the original analysis of the regridded models, presented in Mottram et al (2021) no significant biases were found to have been introduced by the regridding scheme on a continental or bain scale, however, bilinear interpolation does inevitably affect the very local scale. This is explored in relation to the point SMB data comparison shown in Mottram et al. It does not affect or change the results we show here.

10.L50: re-highlight that this is based on the re-gridded versions
Done, we added regridded to the sentences: *"All calculations are derived by subtracting the regridded native mask from the.."*

11. L54: what is the ensemble mean? This is not defined until now – maybe it is included in Mottram21, but values and over what grid should be included here. Need to clearly convey the difference between ensemble mean and 4th column in table 1

See answer to comment #4

12.L55-56: Or parts of the periphery of the ice sheet that has lower SMB are cut off?

No, the difference in SMB arises from the resolution, because the steep coastal orography is better resolved, which leads to more orographic precipitation. We added it to the sentences (Line 74-75 in the revised version):

"...simulation, which is most likely due to the steep coastal orography being better resolved and thus leading to an increased orographic precipitation..."

13. L57-58 incl rightmost column in table 1: I understand from a sea level contribution perspective it is interesting to include the numbers/highlight in impact, but in the introduction, you clearly stated that your ice mask comparison includes shelves etc. I agree that is should be included, it warrants a follow-up, but there is a need to mention this in the Methods – what is defined as grounded ice and are there differences in grounded ice extent in the native ice masks?

Clearly defining the ground ice also becomes an issue in the discussion L80-87(see also below)

See answer to comment #7 here we shortly introduce the grounded values. We have added this to the methods (Line 62-64 in the revised version): "... Zwally et al. (2012) definition. To make the grounded ice sheet values in table 1, we derived the values of the grounded ice sheet, using the data set for grounded ice from Zwally et al. (2012), since not all the native masks had a definition of the grounded ice. This way we insured all the native masks and the common mask were compared equally. In table 2 we had "

14.L83-85: To some extent it makes sense to compare against IMBIE2 but is it a full-on apples-to-apples comparison? Is there "another" common extent for the ground ice only? In which case the comparison makes sense (grounded v grounded), but from reading the manuscript I am not sure where the numbers for the comparison originate from. Please clarify in the methods, results, and here in the discussion

We compare grounded ice sheet SMB with the total AIS mass imbalance as calculated by IMBIE2 ( $109\pm56$  Gt yr -1), which is also only applied over the grounded part of the ice sheet so the comparison is valid. We have rewritten this in the discussion section to make it clearer (Line 111-115 in the revised version).

"Although the change in SMB is relatively small, the total AIS mass budget is close to being in balance, but slightly negative, as determined over grounded ice by IMBIE2. This means that small changes in one of the components, in this case, SMB, can lead to a non-negligible change in the total mass budget of the AIS. We compare the IMBIE2 results over the grounded ice mask, with the Delta SMB in Tab.1, for example, the HIRHAM...."

15. L 85-87. Reads unclear, is something missing. add "."after zero Corrected

- 16. L92: is it also a melt v precip issue? Perhaps worth clarifying? No, it is not due to melt, because there is little to no melt in the data. The data is same as in Mottram et al 2021, here it is seen that the SMB from HIRHAM MetUM and COSMO-CLM does not include melt/runoff, and the SMB for RACMO and MAR does include a runoff component, however these runoff values are very close to zero. Thus, it is not melt/runoff that create the differences in SMB.
- 17.L93: where does 63 come from?

It is the number of different mask combinations when combining the regridded native masks, meaning from the six regridded native masks we can create 63 different ice masks, besides the common mask. To clarify it, this is added to the text:

"In each grid cell between 1 and 6 native masks can have ice in it, this results

**in 63 combinations of model coverage, in addition to the common..."**

18.L94-98: this brings back the point about the origin of each native ice mask. Following this discussion/mentioning of post-creation modification can be discussed/added.

We have rewritten it to (Line 125-132 in the revised version): "These different area coverage combinations around the ice sheet are partly driven by differences in ice masks and partly by differences in resolution. However, we cannot identify any systematic or model-specific biases on a regional scale. The native ice masks vary around the coastline arbitrarily, which is partly due to the time when the ice mask was created and what data are used to create the native ice mask. For example, the HIRHAM5 and MetUM ice masks are created from data collected three decades ago, so there have been multiple calving events since the data collection. The native ice mask from COSMO-CLM2 and MARv3.10 are created from data collected lore recently and in higher resolution. The higher resolution is also a benefit over AP and in coastal areas where there is complex orography, where a higher resolution also can change the orographic precipitation."

19. L105-124: I really like this part. As part of Step 2, I would suggest/recommend proper surface delineation with a specific year associated, e.g. Antarctic summer 2018/19, or what compares well with the remainder datasets, e.g REMA, etc., to ensure a common data platform that can "easily" be updated every X years. I realize that some datasets such as RGI are getting "old", but I will encourage pursuing more recent data/time stamps for the grids, DEMs, etc., and hope that/encourage other data produces will update their data too. That is a good idea to add a time step for when the data is from, within the files, we have added this at the end of this paragraph:

*"Finally, we suggest that it is noted in the mask file, which data sets are used to create it, and when the data sets are last updated"*

- 20.L117: incl. "the" before tool Done
- 21. Table 1 caption: what does "the ensemble mean (from Mottram21)" refer to? I don't follow. I suggest including parentheses around the column headers to

help the reader figuring more clearly out what is part of what

We have rephased the caption to:

"Table 1.From left to right the columns show: RCM name, $\Delta$ area% and  $\Delta$ SMBGtyr-1 between the common mask and the native mask over the ToAIS,  $\Delta$ SMB% is the difference between the  $\Delta$ SMBGtyr-1 and the SMB ensemble mean from Mottram et al. (2021) which is 2329 Gtyr-1, the yearly SMB for the individual models integrated over the common mask with uncertainties of one standard deviation, and finally the $\Delta$ SMBGtyr-1over the Grounded AIS"

- 22. Table 1 SMB column: add uncertainties a key discussion point is also deltaSMB versus uncertainty of each estimate Uncertainties are added
- 23. Table 2 incl caption: add an extra row summarizing the total, even though it is in table 1, and instead of colors then just have deltaSMB in black below the magenta. Adding this will make it easier to compare the basin to the total. Done
- 24. Fig1: Difficult to differentiate between colors. See suggestion above on adding a new observation data to this or a new figure. Also, perhaps include a blow-up of selected areas around the ice sheet showing both "good" and "bad" examples and the grounded ice extent.
  We had changed the colours and added palen showing the REMA mask over the Antarctic Peninsula

**Anonymous Referee #2**

- 25. More information on why there are differences in the various land-ocean masks is warranted. Please see comment below regarding L35 and L95. This is added at the beginning of the Method section, see the answer to comment #1
- 26. As the coastline evolves every year, and will continue to evolve in the future, what recommendations can be made in terms of using a common ice mask?

Both the grounding line and ice shelf extent change through time. Should the ice mask not dynamically evolve both over the reanalysis era and in the future? The authors only discuss creating one, static, ice mask. This in itself will create uncertainty as it will only be "correct" at one point in time. And what point in time should that be?

**That are some good points, below I have answered them point by point:**

a. What recommendations can be made in terms of using a common ice mask?

That the common mask is easily accessible for all and with clear guidelines on how to use it.

b. Should the ice mask not dynamically evolve both over the reanalysis era and in the future?

Ideally, we would of course evolve ice sheet masks through time, but this requires a dynamical ice sheet model to be fully two-way coupled within a regional climate model. The development of regional coupled climate and ice sheet models is still in its infancy and not yet available within the constraints of computing resources. Over the recent past we do not have sufficient observational data to reconstruct multiple ice sheet extents for most of Antarctica.

Even the highest resolution models are still relatively coarse compared to observed changes in ice sheet extent, e.g. the calving of large icebergs from ice shelves. Updates to the ice mask are therefore usually only necessary for long timescale simulations. We have added the following sentence, in the end of the discussion:

"Ideally the ice mask would evolve over time as the continent changes, however, we feel it is important to first standardize the approach to creating and using ice masks. Thus, we suggest that it is noted in the mask file, which data sets are used to create it, and when the data sets are last updated."

c. This in itself will create uncertainty as it will only be "correct" at one point in time. And what point in time should that be?

There is no single point in time when we have a fully complete and accurate ice mask for Antarctica, all datasets are made up of a mosaic of different types of data assembled into a single land surface cover dataset. More recent datasets are generally higher resolution with

better quality data, which is why we suggest using the REMA data to construct a common mask.

- 27.L16: Remove comma after "Although" Done
- 28.L26: Can you provide an approximate grid cell length in km as well?
  Yes, it is approximately 12.5 km, this is added in the sentence *…onto the same grid at 0.11 ∘ (≈12.5 km) resolution*"
- 29.L35-: Can you please provide more information for these various datasets? Over what time period(s) were each developed? How were the masks developed? What does IGBP mean? On line 95, you go on to speculate why there differences may exist in the ice masks, but this is only very briefly discussed, and in my opinion, insufficiently discussed.

This is added at the beginning of the Method section, see the answer to comment #1

Regarding the line 95, we have added more to this, see the answer to comment #18

- 30.L45: change first instance of defined -> define Corrected
- 31.L58: adn -> and Corrected

References

Eidenshink, Jeff C., and John L. Faundeen. "The 1 km AVHRR global land data set: first stages in implementation." *International Journal of Remote Sensing* 15.17 (1994): 3443-3462

---

## Author Response (AR2)

Hansen et al. 2021
https://doi.org/10.5194/tc-2021-317

[Figure]

Reply to reviewer comments on

**"Brief communication: Impact of common ice mask in surface mass balance estimates over the Antarctic ice sheet"**

by

Nicolaj Hansen, Sebastian B. Simonsen, Fredrik Boberg, Christoph Kittel, Andrew Orr, Niels Sourverijns, Melchoir van Wessen and Ruth Mottram

Dear Editor
Dear Dr. Macgregor,
On behalf of my co-authors and I, we would like to thank you for your comments on our manuscript. We believe that your comments have helped to improve the paper.
We have addressed all your comments one by one below, our replies are written in red.

Best regards,
Nicolaj Hansen

1. Because the manuscript is intended as a Brief communication and is presently >12 unformatted pages long, I must point out two things: 1. The new inset in Figure 1 is helpful but enlarges the figure unnecessarily. Please shrink the inset and perhaps reposition SW of Antarctica. 2. The MS is verbose and several statements are repeated nearly verbatim. Please review the MS one last time aiming to increase concision.

   We have taken your suggestion and made Fig 1 smaller and moved panel B. Regarding the second point, we have made the first part of the discussion more concise:

   *"We find the differences between common and native ice mask areas small (<3%), but it alters the SMB by up to 6% over the ToAIS (140.6 Gt yr-1) when compared to the ensemble mean from Mottram et al. (2021). RACMO2.3p2, MARv3.10 andHIRHAM5 0.11 ◦ all have△SMBGtyr-1values close to or larger than their given uncertainties for their respective SMB estimate. This means that the effect of using the common mask in estimating SMB is close to or greater than the standard deviation of annual mean SMB estimates derived from the interannual variability in modelled SMB. We consider the standard deviation to be a minimum estimate of uncertainty within each model with actual uncertainties likely to be considerably larger, but difficult to estimate accurately (Lenaerts et al., 2019). Over the grounded AIS the common mask alters the SMB by up to 102 Gt yr-1,see Table 1. This difference in SMB is close in magnitude to the grounded AIS mass loss of 109±56 Gt yr-1between 1992 and 2017 determined by the second ice sheet mass balance inter-comparison exercise (IMBIE2, Shepherd et al. (2018)), and thereby essentially determining if the AIS is losing or gaining mass. This means that small changes in SMB can lead to a non-negligible change in the total mass budget of the AIS. The model mean of the grounded△SMBGtyr-1is 54.2 Gt yr-1, which would make a sizeable change in the mass balance results, Table 1. Basin 25 has few or no ice shelves, thus it has one of the largest impacts for △SMBGtyr-1 for both the grounded basin (not shown) and when ice shelves are included."*

2. 21: Shouldn't this new enumeration still mention ice masks as a factor driving the SMB differences?

   No, here we are talking about the Mottram et al 2021 study, which used a common ice mask

3. 42 and throughout: Table not Tab.

   Changed

4. 56-57: Bedmap2 as a whole combines several different data sources, but its ice mask doesn't use all of those listed here. See p. 377 of Fretwell et al. (2013). It appears that the first MODIS Mosaic of Antarctica is the underlying source of the ice mask. Not sure if 125 m or 750 m for underlying tracing of

ice-shelf extent. Please check Haran et al. (2005).

This sentence was based on Fretwell et al. (2013) Table 3. Where they give information on the data they have used to derive the surface grid, which is shown in Figure 7 (still  Fretwell et al. (2013)) their surface grid is the same as their ice mask.

5.  67: since not all the native masks explicitly distinguished grounded from floating ice
    Changed

6.  68: In Table 2 we opted
    Changed

7.  90-92: This new sentence is not quite making sense to me. Please revise.
    We have rewritten it to:
    *"Moreover, it is shown how well the common mask agrees to a newly derived ice mask. We compare the common mask to the Reference Elevation Model of Antarctica (REMA, Howat et al.(2019)) mask over the AP, here it is clear that the common mask is smaller than the REMA in most coastal areas around the AP, Fig. 1 panel B."*

8.  99: Not sure what the new phrase is adding to the argument here.
    It is added on the basis on the second main comment from referee 1. To tell that even though COSMO has the smallest relative difference between Δarea % and ΔSMB %, it does not indicate that it is closer to the "true" SMB, solely that it is least affected by the change in ice mask

9.  123: Here I suggest adjusting to "Given the importance of the ice shelves to the dynamics of grounded ice," because the ice shelves themselves are influenced by force dynamics, so the sentence as is seems unclear.
    Good suggestion, we have changed it

10. 125-126: These differences between area change and SMB change are the result
    Changed

11. 134: decades ago, yet there have been
    Changed

12. 139-140: The additional phrase does not flow well with the original sentence.
    The paragraph has been changed to:

[Figure]

*"The common mask is introduced during the post-processing stage after running the RCMs with their native masks. This has the disadvantage that model variables where the fluxes are linked to the orography, such as precipitation, can introduce a bias, if the native mask is located differently in the domain, compared to the common mask. The same orography bias can be true for winds and thus the sublimation rates as well. High precipitation rates are often strongly linked to the steep orography in coastal areas around Antarctica, especially in West Antarctica and on the windward side of the Antarctic Peninsula (basins 24 and 25), which is also where we see the largest differences in $\Delta SMB_{Gtyr-1}$ in Table 2. Comparing the size of the common mask with the REMA mask over the Antarctic Peninsula shows that the common mask is smaller around most of the coastline, Fig.2 panel B."*

13. 156: common tool
    Changed

14. 168: fix tense change (quantify vs. compared)
    It is changed to
    *"We have quantified the importance of the choice of ice mask for the Antarctic domain by comparing six different ice masks from the RCM"*